# Understanding recruitment to a perioperative randomised controlled trial: protocol for a mixed-methods substudy nested within a feasibility trial of octreotide infusion during liver transplantation

Edgar Brodkin ,[1] Ee-Neng Loh,[1] Michael Spiro,[1,2] Vivienne Hannon,[1] Jez Fabes ,[1,3] S Ramani Moonesinghe ,[4] Duncan Wagstaff [4]

For numbered affiliations see end of article.

**Correspondence to**
Dr Duncan Wagstaff;
duncan_wagstaff@yahoo.co.uk

## ABSTRACT

**Introduction** Recruitment to perioperative randomised controlled trials is known to be challenging. Qualitative methods offer insight into barriers and enablers to participation. This is a substudy within a feasibility randomised controlled trial of octreotide infusion during liver transplantation at two National Health Service hospitals, which will evaluate patient and staff experiences of trial processes. By sharing formative understanding from these methods with the trials team we aim to improve staff–patient interactions and hence recruitment rates.

**Methods and analysis** This prospective mixed-methods study will comprise two workstreams. First, after consent to the randomised controlled trial is sought, all patients will be invited to complete a questionnaire to explore their perceptions of the information given to them and motivating factors that influenced their decision to consent or not. Questionnaires will be analysed using descriptive statistics and framework analysis.

If the recruitment:approach ratio drops below a predetermined ratio or if there are any specific recruitment concerns from the trials team, a second workstream involving mixed-methods fieldwork will be implemented. This will involve audiorecording of recruitment consultations and a follow-up semistructured interview to explore patients' perception of their decision-making regarding recruitment. Semistructured interviews will also be conducted with the recruitment team to establish their views about the trial, barriers to recruitment and ways to overcome them. Recruitment consultations will be analysed using Q-QAT methodology and interviews will be analysed using framework analysis. Findings from both workstreams will be formatively fed back to the trials team to enable iterative improvement to recruitment processes.

**Ethics and dissemination** Approval has been granted by Greater Manchester West Research Ethics Committee (ref 20/NW/0071), the Health Research Authority and the local Research and Development offices. A manuscript detailing the summative findings will be submitted to peer-reviewed journals.

## STRENGTHS AND LIMITATIONS OF THIS STUDY

⇒ The use of qualitative methodologies will give us in-depth insight into the barriers and enablers to recruitment to this perioperative randomised controlled trial (RCT).

⇒ Data gathered across two sites will make the findings more generalisable to other centres.

⇒ Formative feedback to the RCT team during ongoing recruitment will allow modification of the recruitment process to improve recruitment rates.

⇒ The role of the two of the researchers (EB and DW) as anaesthetists and members of the RCT trial management group may affect the conduct of the recruitment team interviews, although they will not be performing the interviews themselves.

**Trial registration number** NCT04941911.

## INTRODUCTION

Multicentre, double-blind randomised controlled trials (RCTs) are the methodology of choice for investigating the effectiveness and safety of healthcare interventions.[1] Previous research has shown that recruitment to RCTs can be challenging.[2] Barriers to participation may include, for example, concern over the concept of randomisation, incomplete explanations of trial methodology, or a lack of balance in the way that treatment arms are explained to potential participants by research staff.[2] These difficulties in recruitment can result in failing to start, abandoning or revising target sample sizes of RCTs.[3–5] Furthermore, in multicentre trials, poor or unequal recruitment can negatively impact on staff morale, equality of

workload and trial costs in addition to introducing bias and reducing statistical power.[6]

Previous systematic reviews have shown that methods implemented to improve recruitment to research studies in general have not shown clear benefit. Multicentre trials are also under-represented in this literature.[7] However, Donovan *et al*[6 8] have demonstrated how contemporaneous qualitative research methods can improve the rates of randomisation and informed consent in multicentre RCTs, including in the perioperative setting. They described how with improved understanding of recruitment processes (both 'as planned' and 'as done'), as well as patient perceptions of recruitment, this information can be formatively fed back to trial teams to enable timely adjustments to be made. Rooshenas *et al*[9] have further demonstrated how these interventions can support recruitment in several UK multicentre RCTs. These included one RCT with a placebo arm, such as in our RCT, which can present additional challenges to recruitment such as difficulty for recruiters in articulating the placebo arm.

This mixed-methods substudy is nested within a double-blind randomised, placebo-controlled feasibility trial investigating the use of octreotide infusion during liver transplantation at two National Health Service (NHS) hospitals. Octreotide infusions are currently used during liver transplantation in some centres to potentially reduce bleeding, improve renal outcomes and improve haemodynamic status. However, this practice is based on observational studies. This trial (henceforth referred to as 'the RCT') will be the first to assess the feasibility of randomising patients undergoing liver transplantation to receive either octreotide infusion or placebo. The protocol for the RCT has already been submitted for publication (Manuscript ID: bmjopen-2021-055864.R1).

Consent for participation and recruitment into the RCT will be sought when patients come for an inpatient assessment prior to being placed on the transplant waiting list. Patients already on the waiting list will be contacted by telephone. As there can be a prolonged time period on the transplant waiting list, consent will be confirmed ('enrolment') on admission to hospital for their transplant (online supplemental appendix 1). This strategy has been successfully implemented in comparable interventional studies in this population.[10] The focus of our study will be the initial recruitment stage.

## Aims and objectives

### Aim
To evaluate the barriers and enablers to recruitment to the RCT.

### Objectives
1. To survey patients' reasons and motivation for participation in the RCT using a written questionnaire.
2. In the event of low recruitment rates, to undertake in-depth mixed-methods evaluation of consultation recordings and patient/staff interviews with regard to the recruitment processes.
3. To provide formative and summative feedback to the RCT team, which will enable necessary adjustments to the recruitment process

### Research questions
1. What are the barriers and enablers to patient recruitment to this RCT?
2. Can the recruitment processes be optimised to improve recruitment throughout the feasibility study or for any follow-on substantive trials?

## METHODS AND ANALYSIS
### Study design
This is a nested mixed-methods substudy of a feasibility RCT exploring the use of an octreotide infusion during orthotopic liver transplantation across two centres; the Royal Free Hospital (RFH) and University Hospital Birmingham (UHB). The RFH will be the lead site where the team who led the grant application and study design are based.

This substudy comprises two workstreams: a questionnaire of all patients approached for recruitment to the RCT; and mixed-methods fieldwork. The mixed-methods fieldwork will only be undertaken if the observed recruitment:approach ratio is below predetermined thresholds of <0.3 at the RFH, or <0.15 at University Hospitals Birmingham (UHB), after at least 12 patients at each site, or if the RCT team have specific concerns about recruitment processes. The predetermined thresholds have been chosen to reflect recruitment rates during a previous RCT in this patient population at the RFH,[10] and lower recruitment rates at secondary centres.

Interim findings from the substudy will be fed back to the trial management group (TMG) of the RCT at fortnightly meetings. This will enable us to pick up and respond to themes we identify both during the feasibility study and potentially in our planned full randomised trial and will enable the TMG to make timely adjustments to recruitment processes where needed. These will depend on the issues identified, but may include written guidance, confidential feedback or additional training. Summative findings will be provided to the various stakeholders including the patient representative and the clinical teams.

### Patient and public involvement
The patient representative of the TMG, who has experience with both liver transplantation and recruitment to research studies, was involved in reviewing and refining study design and methodology. Their feedback was used to draft and amend study documentation. They will continue to be involved for the duration of the study and will have input into the dissemination of both the formative and summative findings.

## Eligibility and consent

All patients eligible for the RCT will be eligible for recruitment to the sub-study. Completion of the questionnaire will be taken as implied consent. Participants who are approached for recruitment to the RCT, during their inpatient assessment or via telephone, will be provided with a written patient information sheet (PIS) together with the contact details of the recruitment team (online supplemental appendix 2). They will be given at least 24 hours to review the PIS prior to their recruitment consultation. The substudy will be discussed with them (either in person during the admission for inpatient assessment or by telephone) prior to the recruitment consultation.

If the mixed-methods fieldwork is initiated, all members of the recruitment team and all patients approached for trial participation will be eligible for recruitment. Individuals will only be excluded if they refuse to provide written consent or if they do not speak English. The recruitment team will also be given at least 24 hours to review the PIS prior to their interviews (online supplemental appendix 3).

## Recruitment

All patients approached for consent to the feasibility RCT will be invited to complete the questionnaire after that consultation. The RCT aims to enrol 30 patients within 10 months but will likely have to recruit a far greater number of patients to achieve this as not all patients on the liver transplant waiting list will receive a graft organ within the recruitment time frame (online supplemental appendix 1). Recruitment is due to start from May 2022. There is therefore no specific targeted sample size for the questionnaires; this phase will conclude when recruitment to the RCT ends.

If the mixed-methods fieldwork workstream is triggered, subsequent consecutive patients will be asked for consent (online supplemental appendix 4) to have their recruitment consultation (face to face or telephone) recorded and then be interviewed afterwards. Recruitment will continue until theoretical saturation has been achieved; this is likely to occur after approximately 12 patients have been recruited. Recruiters at both sites will be interviewed until theoretical saturation has been achieved (likely to be six recruiters).

## Data collection

The questionnaire is based on a previously validated questionnaire used in a similar scenario.[11] It explores patients' perceptions of the trial information given to them and their rationale behind agreeing or declining to consent for the RCT. This has been adapted for our study with input from a patient expert on the TMG (online supplemental appendix 5). For the patients contacted via telephone, questionnaires will be completed in hard copy and returned to the study team.

The fieldwork will comprise three different approaches: audio recordings of the recruitment consultation, patient interviews and recruiter interviews.

Audio recordings of face-to-face and telephone recruitment consultations to the RCT will be taken via an encrypted digital voice recorder. Semistructured interviews with patients will be conducted by telephone at least 24 hours after the recruitment consultation. This will be recorded using an encrypted digital voice recorder. The interviews will focus on the patients' perception of the information they were given about the recruitment consultation and the rationale behind their decision to consent to the RCT or not.

Semistructured interviews with members of the research team will be conducted in person or via telephone and recorded on an encrypted digital voice recorder and professionally transcribed.

All interviews will be performed by experienced interviewers who are not members of the trial TMG and do not undertake trial recruitment. As EB and DW are members of the trial TMG they will not be performing the interviews. All recordings will be professionally transcribed. Interview Topic Guides (online supplemental appendix 6 and 7) will be used to ensure consistency and will be updated iteratively based on feedback from the TMG.

## Data analysis

Questionnaires will be analysed formatively every 20 patients approached and summatively at the end of the feasibility study. Responses to closed questions will be summarised using descriptive statistics and thematic analysis for any open-ended answers.

Audio recordings of recruitment consultations will be analysed according to the Quanti-Qualitative Appointment Timing (Q-QAT) methodology:[12] this involves summarising recruitment consultations both quantitatively and qualitatively. The quantitative component records the time taken to present each of the RCT treatments (mean, median range; recruiter; centre), the time taken to explain the design, purpose and procedures of the RCT (mean, median, range, recruiter, centre) and total length of appointment.[13] The qualitative component will thematically analyse the interviews using constant comparative techniques from grounded theory and will use a framework designed to incorporate concepts identified from the relevant literature.

The patient and recruiter interview transcriptions will be imported into the NVivo software package. It will then be analysed using framework analysis to address what the barriers and enablers to recruit to an RCT are and whether the trial documentation and recruitment process can be optimised to aid recruitment. We will explore additional themes as they emerge. A codebook will be developed to enable team-based analysis. The researchers will engage in a continuous process of reflexivity by documenting their own assumptions, viewpoints and impacts.

As delineated in Rooshenas et al,[14] a key set of recruitment issues will be devised, triangulating the data from the aforementioned analyses and also quantitative data from the screening log. These will be presented to the chief investigator and at the fortnightly TMG meetings.

## ETHICS AND DISSEMINATION

This study is sponsored by the UCL Joint Research Office (Reference number 125176). Ethical approval has been granted by the Greater Manchester West Research Ethics Committee (reference 20/NW/0071). The Health Research Authority have granted permission for the research to be conducted at the two NHS sites.

The substudy has been prospectively registered with the Study Within A Trial (SWAT) database (reference SWAT 152).[12]

All investigators and trial site staff will comply with the requirements of General Data Protection Regulation with regard to the collection, storage, processing and disclosure of personal information and will uphold the Data Protection Act's core principles. A manuscript detailing the summative findings will be submitted to peer-reviewed journals for publication.

**Author affiliations**
[1]Department of Anaesthesia, Royal Free London NHS Foundation Trust, London, UK
[2]Department of Surgical Biotechnology, Division of Surgery and Interventional Science, University College London, London, UK
[3]University of Plymouth, Peninsula Medical School, Plymouth, UK
[4]Centre for Perioperative Medicine, Research Department of Targeted Intervention, University College London, London, UK

**Contributors** DW, SRM and JF designed the study. EB and DW drafted this manuscript. EB, VH, MS and E-NL will conduct screening and data collection. Analysis will be performed by DW. All authors reviewed this manuscript for intellectual content and approved the final version.

**Funding** This work was supported by the National Institute for Health research Award ID PB-PG-0817-20023.

**Competing interests** None declared.

**Patient and public involvement** Patients and/or the public were involved in the design, or conduct, or reporting, or dissemination plans of this research. Refer to the Methods section for further details.

**Patient consent for publication** Not applicable.

**Provenance and peer review** Not commissioned; externally peer reviewed.

**ORCID iDs**
Edgar Brodkin http://orcid.org/0000-0001-8764-0390
Jez Fabes http://orcid.org/0000-0003-1111-5973
S Ramani Moonesinghe http://orcid.org/0000-0002-6730-5824
Duncan Wagstaff http://orcid.org/0000-0001-9472-1578

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
