## [Reviewer comments · BMJ Open]

ARTICLE DETAILS

TITLE (PROVISIONAL)	Understanding recruitment to a perioperative randomised controlled trial: protocol for a mixed-methods sub-study nested within a feasibility trial of octreotide infusion during liver transplantation
AUTHORS	Brodkin, Edgar; Loh, Ee-Neng; Spiro, Michael; Hannon, Vivienne; Fabes, Jez; Moonesinghe, S. Ramani; Wagstaff, Duncan

VERSION 1 – REVIEW

REVIEWER	Sahmeddini, Mohammad Ali Shiraz University of Medical Sciences
REVIEW RETURNED	04-Mar-2022

GENERAL COMMENTS	Thank you for your submitting manuscript entitled: Understanding recruitment to a perioperative randomized controlled trial: protocol for a mixed methods sub-study nested within a feasibility trial of octreotide infusion during liver transplantation to BMJ. I read this amazing mixed methods research manuscript about challenges during recruitment to perioperative randomized controlled trials. Please clarify the date of this study in the method section.
---

REVIEWER	Jepson, Marcus University of Bristol School of Social and Community Medicine
REVIEW RETURNED	31-Mar-2022

GENERAL COMMENTS	This paper is a protocol for a mix-methods sub-study in a liver transplantation feasibility trial. The introduction sets the context for the study clearly enough. Perhaps also consider inclusion of Rooshenas et al paper (details at foot of the page), which describes the impact of the intervention described in the Donovan citations (6,8) . Given the placebo comparator proposed in the RCT, consider mentioning the particular challenges of recruiting to these sorts of trials. Aims, Objectives and research question all seem clear to me/ Methods are also well described and clear enough to follow. The authors propose using the numbers approached as the denominator in deciding whether to implement the mixed methods study (although I have more to say about this in the recruitment section). I would recommend using numbers of eligible patients in place of numbers approached. We know from previous work that often patients fall off the pathway before being approached, sometimes as a result of recruiter equipoise issues. Eligibility and consent all seems clear
---

	Recruitment The aim of the feasibility is to enrol 30 patients in 10 months. My concern about only implementing the mixed methods workstream if numbers fall below a certain threshold (see above) is that within this short timeframe it may be difficult to actually collect much data. I would propose revising the protocol and including it from the start. A further justification for doing so is that if recruitment rates are high, having these data on how the RCT is described and received will provide insights into successful recruitment practices that could then be applied into a full trial (if it proceeds). Data collection Interviews with patients and recruiters are proposed to be by telephone. A minor point, but you could also offer online interviews (ie zoom) as some patients / staff may prefer this. On P9, I12, the authors state that "All interviews will be performed by experienced interviewers who are not members of the trial TMG and do not undertake trial recruitment" but in the article summary (p4, I18) they state that the researchers ARE members of the TMG. I don't have a problem if they are/are not, but one of the two statements should probably be changed to be consistent. Analysis What is proposed seems reasonable. Perhaps give some consideration to how the different data sources will be combined (or not) in terms of providing feedback. (see also, Rooshenas, triangulation paper, link below) It's not clear from this protocol how the findings from the different data sources will be used to feedback to the TMG team, and how this might then influence/change things. I wonder whether this could be expanded on? Rooshenas, L., By-Band-Sleeve study group, CSAW study group, HAND-1 study group, OPTIMA prelim study group, the Romio feasibility study group, Scott, L. J., Blazeby, J. M., Rogers, C. A., Tilling, K. M., Husbands, S., Conefrey, C., Mills, N., Metcalfe, C., Carr, A. J., Beard, D., Davis, T., Paramasivan, S., Jepson, M., ... Macpherson, I. E. (2019). The QuinteT Recruitment Intervention supported five randomized trials to recruit to target: a mixed-methods evaluation. Journal of Clinical Epidemiology, 106, 108-120. https://doi.org/10.1016/j.jclinepi.2018.10.004 Rooshenas, L., Paramasivan, S., Jepson, M., & Donovan, J. (2019). Intensive Triangulation of Qualitative Research and Quantitative Data to Improve Recruitment to Randomized Trials: The QuinteT Approach. Qualitative Health Research, 29(5), 672-679. https://doi.org/10.1177/1049732319828693
--	--

VERSION 1 – AUTHOR RESPONSE

Responses to peer review comments

Reviewer: 1

Dr. Mohammad Ali Sahmeddini, Shiraz University of Medical Sciences

Comments to the Author:

Dear colleagues

Thank you for your submitting manuscript entitled: Understanding recruitment to a perioperative randomized controlled trial: protocol for a mixed methods sub-study nested within a feasibility trial of octreotide infusion during liver transplantation to BMJ. I read this amazing mixed methods research manuscript about challenges during recruitment to perioperative randomized controlled trials. Please clarify the date of this study in the method section.

Best regards.

Many thanks for your comments. We have now included the proposed start date of the study in the method section.

Reviewer: 2

Dr. Marcus Jepson, University of Bristol School of Social and Community Medicine

Comments to the Author:

This paper is a protocol for a mix-methods sub-study in a liver transplantation feasibility trial. The introduction sets the context for the study clearly enough. Perhaps also consider inclusion of Rooshenas et al paper (details at foot of the page), which describes the impact of the intervention described in the Donovan citations (6,8) . Given the placebo comparator proposed in the RCT, consider mentioning the particular challenges of recruiting to these sorts of trials.

Many thanks for your comments. We have included the very interesting Rooshenas et al paper you referenced and have also made mention of the placebo comparator in the RCT.

Aims, Objectives and research question all seem clear to me/

Methods are also well described and clear enough to follow. The authors propose using the numbers approached as the denominator in deciding whether to implement the mixed methods study (although I have more to say about this in the recruitment section). I would recommend using numbers of eligible patients in place of numbers approached. We know from previous work that often patients fall off the pathway before being approached, sometimes as a result of recruiter equipoise issues.

While we definitely take this point on board, we are constrained by the protocol of the main study, which stipulates the ratio of feasibility as that of approached patients being the denominator. A screening log has been constructed for the main study, which will record all eligible patient as well as approached patients, so this is something that will be reviewed.

Eligibility and consent all seems clear

Recruitment

The aim of the feasibility is to enrol 30 patients in 10 months. My concern about only implementing the mixed methods workstream if numbers fall below a certain threshold (see above) is that within this short timeframe it may be difficult to actually collect much data. I would propose revising the protocol and including it from the start. A further justification for doing so is that if recruitment rates are high, having these data on how the RCT is described and received will provide insights into successful recruitment practices that could then be applied into a full trial (if it proceeds).

We also take this very valid point on board. The study was designed to be very pragmatic, with the more labour-intensive qualitative work only triggered if there were concerns regarding recruitment rates. However, we will be collecting the questionnaire data from the start and we did make a protocol amendment to give us the ability to trigger the fieldwork stream if there are any concerns about recruitment processes and not solely based on recruitment ratios. We also feel that if there are

recruitment problems, we will find them early on when contacting patients already on the waiting list. While these do not address your point on providing insights into successful recruitment practices, we feel this strikes the balance between pragmatism and collecting additional data.

Data collection

Interviews with patients and recruiters are proposed to be by telephone. A minor point, but you could also offer online interviews (ie zoom) as some patients / staff may prefer this.

We acknowledge that this does sound like a good idea, but the protocol only makes mention of audio recording and sadly the protocol has been finalised with recruitment due to commence very soon.

On P9, I12, the authors state that “All interviews will be performed by experienced interviewers who are not members of the trial TMG and do not undertake trial recruitment” but in the article summary (p4, I18) they state that the researchers ARE members of the TMG. I don't have a problem if they are/are not, but one of the two statements should probably be changed to be consistent.

Thank you for pointing this out. We have amended the manuscript to clarify that while the researchers for the qualitative substudy (EB and DW) are part of the TMG, the patient and staff interviews will be performed by others, who are not part of the TMG.

Analysis

What is proposed seems reasonable. Perhaps give some consideration to how the different data sources will be combined (or not) in terms of providing feedback. (see also, Rooshenas, triangulation paper, link below).

We have given this further consideration and have adjusted the manuscript to reference the Rooshenas triangulation paper and some of the different combined data sources.

It's not clear from this protocol how the findings from the different data sources will be used to feedback to the TMG team, and how this might then influence/change things. I wonder whether this could be expanded on?

We have amended the manuscript to expand on how the findings will be used to give feedback to the TMG and then make changes. As we don't yet know what issues might be identified, we have just included some examples based on issues we anticipate could be an issue based on some of the literature.

Rooshenas, L., By-Band-Sleeve study group, CSAW study group, HAND-1 study group, OPTIMA prelim study group, the Romio feasibility study group, Scott, L. J., Blazeby, J. M., Rogers, C. A., Tilling, K. M., Husbands, S., Conefrey, C., Mills, N., Metcalfe, C., Carr, A. J., Beard, D., Davis, T., Paramasivan, S., Jepson, M., ... Macpherson, I. E. (2019). The QuinteT Recruitment Intervention supported five randomized trials to recruit to target: a mixed-methods evaluation. *Journal of Clinical Epidemiology*, 106, 108-120. <https://doi.org/10.1016/j.jclinepi.2018.10.004>

Rooshenas, L., Paramasivan, S., Jepson, M., & Donovan, J. (2019). Intensive Triangulation of Qualitative Research and Quantitative Data to Improve Recruitment to Randomized Trials: The QuinteT Approach. *Qualitative Health Research*, 29(5), 672-679. <https://doi.org/10.1177/1049732319828693>

As discussed above, we have reviewed these references and included them in our manuscript.

VERSION 2 – REVIEW

REVIEWER	Jepson, Marcus University of Bristol School of Social and Community Medicine
REVIEW RETURNED	09-May-2022
GENERAL COMMENTS	I am satisfied that the authors have taken on board my comments.